# Polymer Composite Materials Fiber-Reinforced for the Reinforcement/Repair of Concrete Structures

**DOI:** 10.3390/polym12092058

**Published:** 2020-09-10

**Authors:** George Soupionis, Pantelitsa Georgiou, Loukas Zoumpoulakis

**Affiliations:** Department of Chemical Engineering, National Technical University of Athens, 9 Heroon Polytechniou Str., Zografou Campus, 15773 Athens, Greece; litsacyp@central.ntua.gr (P.G.); lzoubou@chemeng.ntua.gr (L.Z.)

**Keywords:** polymer composites, carbon fiber, reinforced concrete structures, artificial aging

## Abstract

The present paper deals with the use of polymeric matrix composite materials reinforced with carbon fiber as concrete shear reinforcement materials. Accordingly, cement specimens were manufactured and coated with various types of carbon fabrics and epoxy resin in liquid and solid form (paste). Additionally, composite materials of epoxy resin matrix reinforced with carbon fiber fabrics were manufactured. In all the specimens, the mechanical properties were estimated; the cement samples coated with composite materials of epoxy resin matrix reinforced with carbon fiber fabrics were tested for compressive strength, while the other specimens were tested for shear and bending strength. The specimens were subjected to artificial aging through heat treatment for 8, 12 and 16 days. During the process of artificial aging, the temperature in the chamber reached the range of 65–75 °C. These composite materials exhibited high mechanical properties combined with adaptability. Both an external deterioration of the materials as well as a reduction in mechanical properties during their artificial aging heat treatment were observed. This was shown in the specimens that were not subjected to artificial aging, with an applied compression strength of 74 MPa, and after the artificial aging, there was a decrease of ~7%, with the compression strength being reduced to 68 MPa.

## 1. Introduction

Polymer matrix composite materials are widespread in many kinds of applications due to the combination of the excellent properties they possess [1]. The most common composite materials are those of a thermosetting polymer matrix (epoxy or phenolic) reinforced with glass fibers or carbon fibers. In this form, they are applied in many sectors [1,2,3], such as shipbuilding, automobiles, construction, etc. For applications with high requirements where a high modulus of elasticity is demanded [4,5]—e.g., in the aerospace–automotive industries and in sporting goods—carbon fibers or aramid fibers are used since both fiber categories exhibit a high modulus of elasticity in addition to high tensile strength.

The concrete structures that are exposed to earthquakes or extreme weather conditions are the most vulnerable structures, hence the vital need for their repair. This is especially important for the non-reinforced structures (concrete and brick masonry), which are the areas that undergo more stress. Since they constitute a very promising field, cement matrix materials reinforced with composite materials have been widely studied over the last two decades [6,7,8,9]. This is important since the use of brick masonry that carries force loads is not standard in a number of countries. For example, in some countries, brick masonry is used only as a covering surface filling the space between concrete columns (hollow bricks). Otherwise, the reinforcement of brick masonry constitutes a specific sub-field that has been studied thoroughly, yielding, in that respect, effective solutions [8], but none has been standardized to this day. Such composite materials find application in construction due to their exceptional shear strength and ease of application in complicated structures. These are the main reasons for the choice of this type of composite material for this study [10]. The application of composite materials in such constructions primarily concerns the main structure and, specifically, the columns [9,11,12].

The fact that the concrete structures accept the loads vertically to the small surface of the structure makes necessary the reinforcement of the rest of the surfaces. In particular, it is necessary to extend the life of concrete structures with the reinforcement of the main structure, not only of the columns but also of the concrete masonry (wall). The outer reinforcements of concrete structures are important in two ways:It is without question that this type of reinforcement cannot remove the inherent inner imperfections that might lie hidden within these concrete structures—and at the same time, they are difficult to identify [9]. However, it is exactly because of these difficulties in total repair that the outer reinforcements need detailed study with respect to their role in extending the body lifespan and also delaying further inner debonding.Outer reinforcements protect the whole structure from the effects of external factors that contribute to their deterioration.

The aim of this study is to contribute to the expansion of the experimental literature on the subject of the compression strength of cementitious structures using composite materials as external reinforcement. Additionally, it is essential to examine how the climate (artificial aging) affects the mechanical properties of concrete that has been reinforced with composite material. For this purpose, cement specimens were manufactured according to EN 196-1: 1995 and were coated with carbon fiber fabrics (unidimensional and two-dimensional) and epoxy resin. Although it is used in complex shapes, the two-dimensional fabric is not commonly utilized for the reinforcement of this type of structure. Additionally, composite materials were manufactured with epoxy resin (liquid and paste) as a matrix and carbon fiber fabrics as a reinforcement, to estimate their mechanical properties and, in particular, their flexural and shear strength. For comparison reasons, the compressive strength was estimated in cement samples, both reinforced and non-reinforced ones. Furthermore, cement samples reinforced with carbon fiber and epoxy resin were subjected to artificial aging to investigate to what extent the mechanical properties were affected over time in healthy but also vulnerable surfaces.

## 2. Materials and Methods

Epoxy resins, which have been used as a matrix for the manufacturing of composites, in liquid and paste form, are readily available in the market. Specifically, liquid form epoxy resin (SINTECNO) is a low viscosity solvent-free impregnation resin and is used in the application of reinforcing fabrics. For the better application of liquid epoxy resin to reinforcing fabrics, a resin of a similar base (SINTECNO) is used as an adhesive (primer). The paste epoxy resin (SIKA) is a high strength resin and is mainly applied to reinforcing fabrics.

For the reinforcement, the carbon fiber fabrics used are of two types, unidimensional and two-dimensional (twill) (Figure 1a,b). The twill pattern is elastic and good for use with complex shapes because its weave is looser, as shown in Figure 1c. In this type of concrete structure, it is not generally common in the literature to use twill pattern carbon fiber fabric for reinforcement. Thus, the twill pattern was chosen for this application because it can be used with complex shapes. While the unidimensional carbon fiber fabric has a mono-fiber number of 3000 and a density of 1.8 g/cm^3^, the two-dimension carbon fiber fabric is twill knit. It has a mono-fiber number of 3000 and a density of 1.8 g/cm^3^.

The cement specimens were manufactured with Portland-type cement and sand following the EN196-1 standard, with the following proportions: cement, 450 g (±2); sand, 1350 g (±5); and water, 225 g (±1), using the cement mortar presented in Figure 2a. After curing, the cement specimens were placed in water for 28 days (Figure 2b). After then, and when they were completely dried, the surfaces were mechanically rubbed to smooth surfaces for the better adhesion of the resins. Besides, their edges were also smoothed for the better application of carbon fabric, since the edges suffer the greatest tension/pressure (Figure 3a). Any defective sites on the surfaces of the specimens were treated with special paste epoxy resin (SINTECNO) with additives (quartz powder)—for smoothing the surface.

Once the surfaces of the specimens had been properly prepared, they were coated with the composite material.

The specimens, where liquid resin was used, were firstly treated with a primer for better adhesion of the fabric to the cement surface (Figure 3b). The cement specimens were reinforced peripherally to their four sides (Figure 4a), with two layers of the composite material. This reinforcement method was chosen in order to increase the shear capacity of the concrete columns, to enhance their ductility and to improve the confinement level of the concrete during mechanical stress [13,14].

The unidimensional carbon fiber fabric was used to wrap the cement specimen, with the direction of the fibers placed vertically to the compression force. As for the two-dimensional fabric, one direction of its fibers was parallel and the other one was vertical to the compression force, as shown in Figure 1. Moreover, paste epoxy resin was easier to apply during the manufacturing process. After the cement specimens were coated with the composite material, specimens from all cases were selected and subjected to heat treatment (aging) at 65–75 °C for 8, 12 and 16 days, for comparison purposes. The compressive strength was measured in all the cement specimens.

To measure their bending and shear strength, the composites were made with both liquid and paste resin with unidimensional and two-dimensional (twill) fabric. All the composites were manufactured using the hand-layup method [8]. Figure 4b shows the dimensions of the specimens for the bending and shear strength measurements.

The calculation of the carbon fiber fabric layers for the manufacture of the composite materials was in accordance with the thicknesses of the fabrics. For the unidimensional fabric, the thickness was 0.630 mm, and for the two-dimensional (twill) fabric, the thickness was 0.285 mm. Thus, to manufacture composite materials with a 3 mm thickness, for the unidimensional fabric are required five (5) layers (according to ASTM D7617), which weigh 358 g, and for the two-dimensional (twill) fabric are required eleven (11) layers (according to ASTM D7617), which weigh 405 g. Furthermore, to manufacture the composite materials in the case of the unidimensional fabric, the epoxy resin that was used weighed 358 g, and that used for the two-dimensional (twill) fabric weighed 405 g. The volumetric ratio of fiber/resin for the composite material was 53% (*v/v*), and for the reinforced cement specimens, the fiber/volume ratio was 3% (*v/v*) (for two layers of carbon fabric). The specimens that were manufactured in the present work are depicted in Figure 5a. Those specimens suitable for the mechanical tests are shown in Figure 5b.

## 3. Mechanical Properties

The mechanical properties of the composite materials (Figure 5a), specifically, the bending strength and shear strength, were determined according to the three-point method based on ASTM D790-71, ASTM D7617 and D2344-65T (BS EN ISO 14125: 1998) (Figure 6). Based on the standards for overall bending tests, the support span should be 16 (tolerance ±) times the thickness of the specimen (in this work, 5 cm), and the speed, no higher than 2 mm/min. For the shear tests, the support span should be eight (tolerance ±) times the thickness (in this work, 2 cm), and the speed, no higher than 2 mm/min. For both of the tests, the specimen is placed horizontally on a support span, and the load is applied to the center by the loading nose of a special dynamometer, which, by applying pressure (1 mm/min), measures the resulting deflection in proportional indication. The result corresponds to a force in a table given by the instrument manufacturer, and then, the bending strength *σ_b_* (MPa) and the shear strength *τ_b_* (MPa) are calculated according to Equations (1) and (2).
(1)$$\sigma_{b}=\frac{3P_{max}l_{s}}{2bd^{2}}$$
(2)$$\tau_{b}=0.75\frac{P_{max}}{bd}$$
*P*_max_ is the maximum load as the specimen breaks (N), *l_s_* is the test length (m), *b* is the width (m) and *d* is the thickness (m).

The compressive strength was measured for cement specimens wrapped in carbon fabric and epoxy resin as the matrices, following the EN 196-1:1995 standard. The tests were carried out on an INSTRON press of HTM-HYD 300 kN (Figure 7). The result gives the power in kN, and the compressive strength *σ_θ_* (Mpa) is calculated according to Equation (3).
(3)$$\sigma_{\theta }=\frac{F}{A}$$
*F* is the maximum load when the specimen breaks (kN), and *A* is the specimen surface area contacting the press (mm^2^).

## 4. Artificial Aging

The process of artificial aging was carried out in a SUNSET CPS ATLAS booth (Figure 8), and the specimens were exposed to UV radiation at 550 W/m^2^ (295–800 nm). Due to the large short wavelength radiation flux, UVB sources cause faster degradation than solar radiation, making UVB sources ideal for accelerated material strength tests, such as for polymers that are mainly sensitive to shorter wavelengths [15,16]. During the process of artificial aging, the temperature in the chamber reaches the range of 65–75 °C.

## 5. Results and Discussion

To manufacture the cement and composite specimens, epoxy resin and unidimensional and two-dimensional carbon fiber fabrics were used, as shown in Table 1.

Figure 9 shows specimens that were tested for bending and shear strength. All the specimens (of all the cases) that were tested for shear strength broke completely. By contrast, those that were tested for bending strength did bend, but they did not break. This is because in bending strength tests, the fabric receives the load, preventing the specimen from breaking.

Figure 10 shows the bending strengths and Figure 11 shows the shear strengths of the composite materials with paste and liquid epoxy resin as matrices and reinforced with unidimensional and two-dimensional carbon fiber fabrics. It is apparent from Figure 10 that the composite materials with unidimensional (Od) carbon fabrics outperformed, in terms of bending strength, the two-dimensional (Twill) carbon fabrics by ~30%. As for the matrix, the liquid form epoxy resin exhibited greater bending strength, which was 405.13 MPa. As shown in Figure 11, the shear strength was approximately the same in all cases at around 30 MPa. Specifically, the specimens that were reinforced with unidirectional fabrics exhibited slightly higher shear strength (33 MPa).

As shown in Figure 12, the breaking of the specimens in all cases was as expected [17,18], perpendicular to the dimension of the force exerted on them. In all three cases (a, b and c), the fracture was at the edge of the specimens, where it was the most sensitive point. What was observed during the experiments was that the specimens that had been reinforced with carbon fiber cloth with liquid epoxy resin, in the majority, did not show any rupture on the surface of the composite material.

It is evident in Figure 13 that in the case of cement specimens reinforced with the composite material of the two-dimensional carbon fabric and paste resin (Cem_twill_paste), the compressive strength increased by 42%, reaching a value of 74 MPa, compared to that of those without reinforcement. In all other cases, the compressive strength increased by ~20%. As far as the matrix is concerned, paste epoxy resin performed better in all cases and thus increased the durability of the specimens.

The effects of artificial aging in all cases are obvious (Figure 14), while all specimens were superficially altered. In the paste resin specimens, it is evident that the resin had created a layer of dust on the surface of the specimens, and the liquid resin specimens look burned. These show the aging of the resin layer promoted by its oxidation [19].

Figure 15 shows the specimens that were subjected to compression strength measurements after being exposed to UV radiation. In these specimens, the breaks were near their edges, too. This shows that their exposure to UV radiation did not affect the way in which the composite material received the loads. The surface degradation was greater in the case of specimens reinforced with epoxy paste, regardless of the means of reinforcement (one dimensional or twill carbon fiber sheet). Despite the wide surface degradation of the epoxy-reinforced specimens, as shown in Figure 15a, their compressive strength remained higher (see below Figure 16).

It was observed during the compressive strength measurement procedure in the case of the paste resin specimens that they peeled at very low force.

Figure 16 shows the compressive strength of the specimens subjected to artificial aging heat treatment. The decrease in strength is evident in all cases, with a ~10% difference from those that did not undergo artificial aging for the first 8 days of exposure. With prolonged exposure of the specimens, the strength, as shown in Figure 16, decreased. Regarding the 16-day exposure and in the case of the cementitious specimens coated with liquid epoxy resin matrix, the compressive strength was reduced by 30% compared to that of those that were not subjected to artificial aging. However, for the paste epoxy resin specimens, the strength reduction rates were different per case. For the specimens with unidimensional carbon fabrics (Cem_od_paste), the reduction rate was 20%, with the strength reaching 46.3 MPa, while in the case of the two-dimensional fabric (Cem_twill_paste), the percentage was greater, at 40%, with the strength reaching 44.9 MPa.

Based on Figure 17, the loss of mass occurred during the artificial aging heat treatment in addition to the decrease in compressive strength. In the first days of exposure, the resin on the specimens emitted a strong smell of volatiles due to the evaporation of superfluid solvents, i.e., MEK (methyl ethyl ketone), which is present in the hardener of the resin. Those volatiles constitute the 3% to 6% mass loss, and the remaining (~2%) is water from the cement, as can be observed in Figure 17. The cement specimen had the smallest loss of mass, due to the water evaporation, as expected [20]. The greatest loss was observed in the specimens coated with liquid resin, being 8%, from which it is easier for solvents to escape. All the specimens show a gradual increase in loss of mass with increasing exposure time, up to a limiting value near sixteen days.

## 6. Conclusions

This paper presents an experimental and theoretical investigation on the compression behavior of reinforced small-scale cement beams with CFRP reinforcement after they have been heat-treated (artificial aging). The main findings drawn are as follows:Unidimensional carbon fiber fabrics as reinforcement provide better bending strength than the two-dimensional ones with twill weaving, in all cases, independent of the form of the epoxy matrix.The maximum flexural strength was found in the composite material of liquid epoxy resin reinforced with unidimensional carbon fiber fabric, due to the better penetration of the resin into the fibers.It was found that the shear strength of composite materials does not change significantly depending on the case, with the cases of composite materials reinforced with unidimensional fabrics being slightly higher in strength.An increase in compressive strength of 42% was observed in the case of a cement specimen reinforced with composite material of paste epoxy resin with a two-dimensional carbon fiber fabric (Cem_twill_paste). While the choice of the twill pattern indeed generated better results, the role of the matrix of the composite material cannot be overstated.Additionally, it was observed that paste epoxy resin was easier to apply to the specimen with carbon fabrics. However, it is worth noting that the two-dimensional carbon fiber fabric adhered better to the specimen.Artificial aging, the procedure of heat treatment, induces a significant reduction in the mechanical properties of composite materials of up to 40%. This is also evident in paste resin specimens where, by the end of artificial aging, the resin had created a layer of dust on the surface of the specimens. This points to the resin oxidation and the evaporation of superfluid solvents in the hardener. The composite materials with life-extending anti-UV epoxy resins are expected to maintain their high mechanical properties.

This study clearly reveals that the composite material of paste epoxy resin as a matrix and twill fabric as reinforcement produces better results in reinforcing cementitious specimens. Of all the specimens that were subjected to artificial aging, the aforementioned ones proved to exhibit the best mechanical properties.

## Figures and Tables

**Figure 1 polymers-12-02058-f001:**
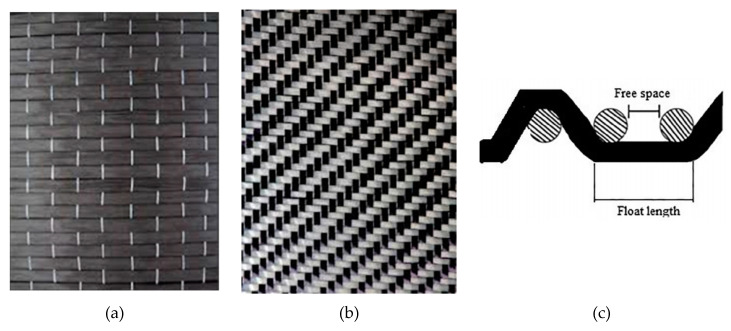
Carbon fiber sheet: (**a**) unidimensional and (**b**) two dimensional twill (**c**) weaving of the twill pattern.

**Figure 2 polymers-12-02058-f002:**
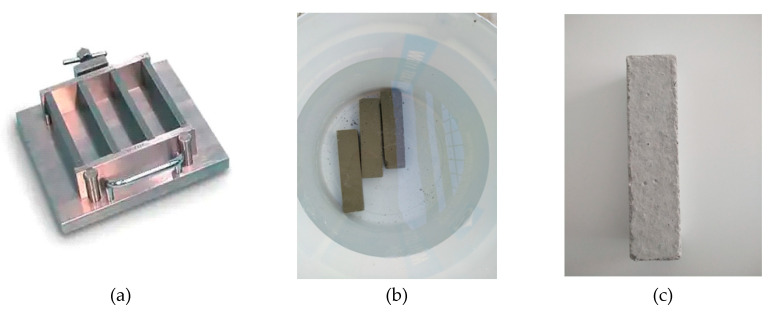
(**a**) Three-gang mold for 40 × 40 × 160 mm^3^ cement mortar; (**b**) Cement specimens placed in the water for 28 days; (**c**) Cement specimen.

**Figure 3 polymers-12-02058-f003:**
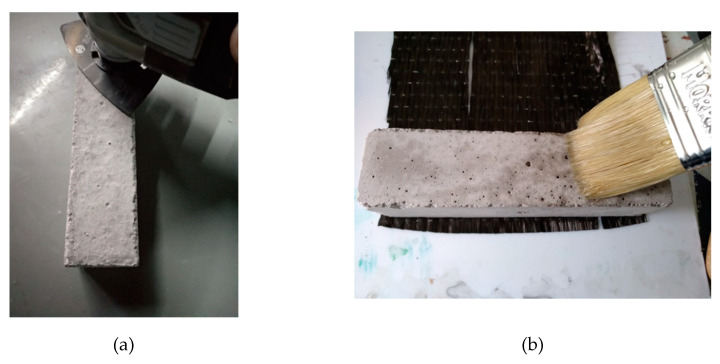
(**a**) Mechanical smoothing of the specimen surfaces; (**b**) Priming the surfaces for better adhesion of the fabric.

**Figure 4 polymers-12-02058-f004:**
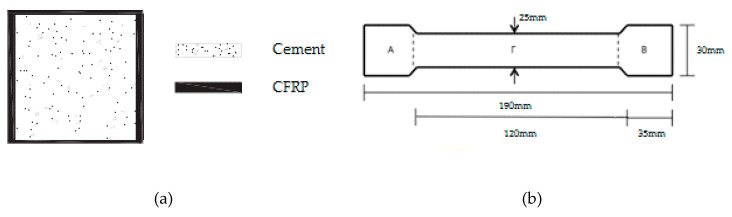
(**a**) Cement specimens reinforced peripherally with CFRP; (**b**) Specimen dimensions for the bending and shear strength measurement.

**Figure 5 polymers-12-02058-f005:**
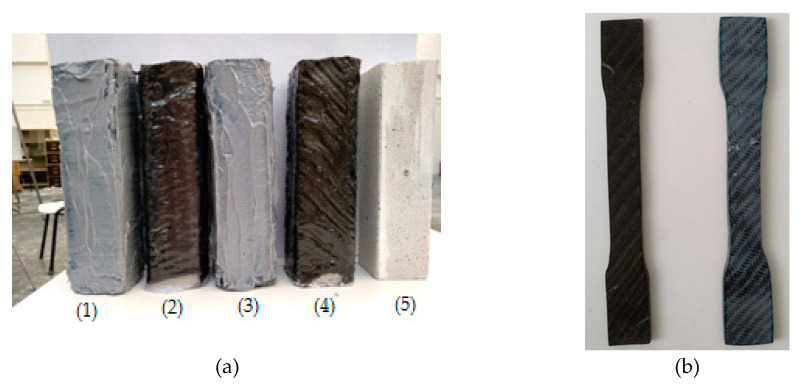
(**a**) All cases of cement specimens that were manufactured: (1) paste epoxy with unidimensional fabric, (2) liquid epoxy with unidimensional fabric, (3) paste epoxy with twill fabric, (4) liquid epoxy with twill fabric, (5) cement specimen; (**b**) Specimens that were manufactured for the bending and shear strength measurements (case of twill fabric).

**Figure 6 polymers-12-02058-f006:**
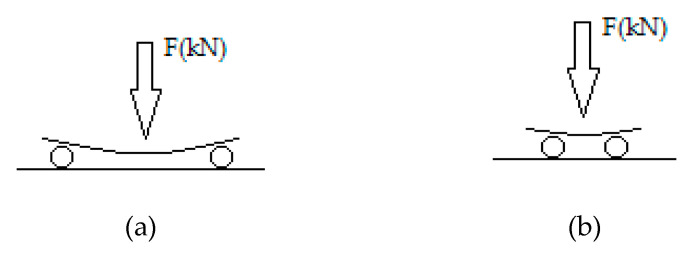
(**a**) Three-point scheme for the bending strength measurement; (**b**) Three-point scheme for the shear strength measurement.

**Figure 7 polymers-12-02058-f007:**
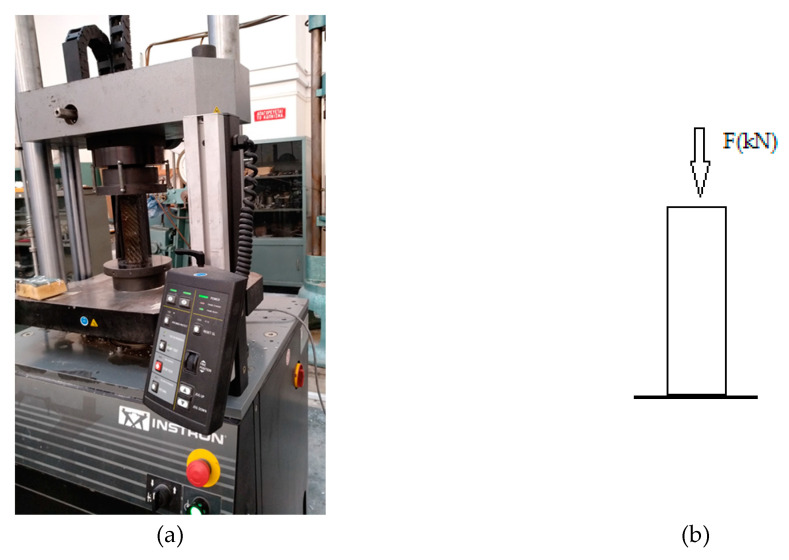
(**a**) INSTRON press of HTM-HYD 300 kN during the compression tests; (**b**) Representation of compression measurement.

**Figure 8 polymers-12-02058-f008:**
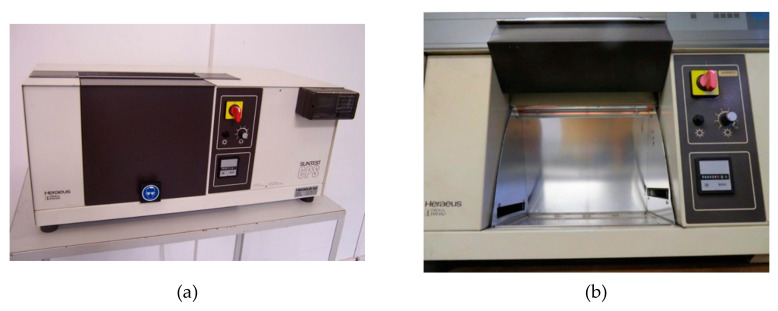
(**a**) SUNSET CPS ATLAS used for the artificial aging; (**b**) Booth where the specimens were exposed to UV radiation at 550 W/m^2^ (295–800 nm).

**Figure 9 polymers-12-02058-f009:**
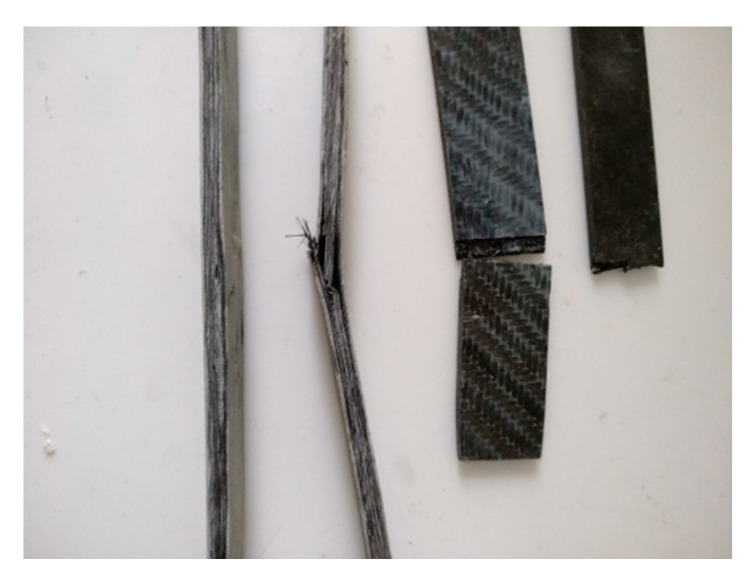
Specimens tested in bending and shear strength.

**Figure 10 polymers-12-02058-f010:**
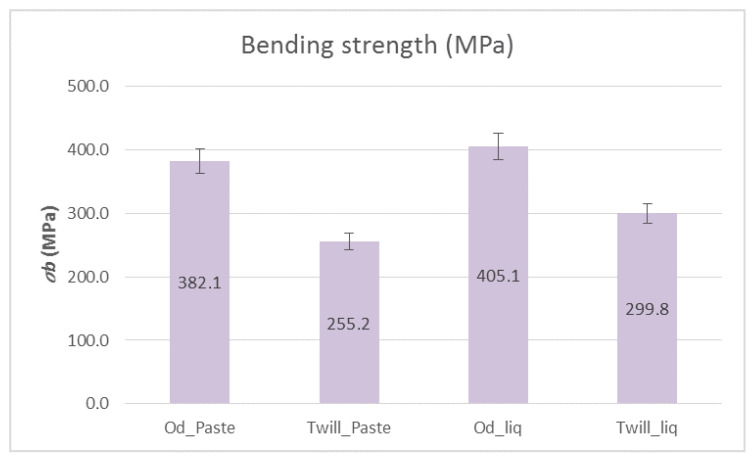
Bending strength of paste and liquid epoxy resin specimens reinforced with carbon fiber fabrics. Variation of values, ±5%.

**Figure 11 polymers-12-02058-f011:**
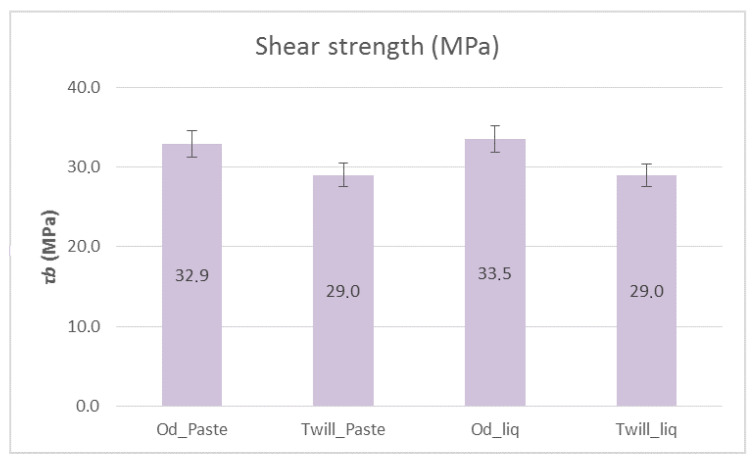
Shear strength of paste and liquid epoxy resin specimens reinforced with carbon fiber fabrics. Variation of values, ±5%.

**Figure 12 polymers-12-02058-f012:**
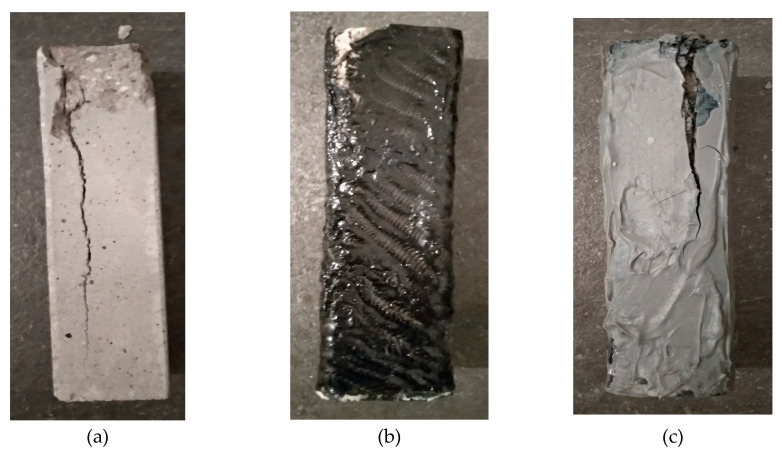
(**a**) Compressed cement specimen without reinforcement; (**b**) Compressed cement specimen reinforced with liquid epoxy and carbon fiber sheet; (**c**) Compressed cement specimen with paste epoxy and carbon fiber sheet.

**Figure 13 polymers-12-02058-f013:**
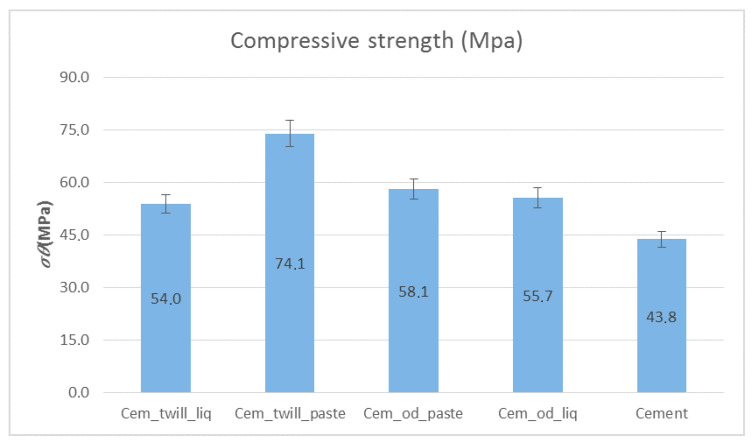
Compressive strength of cementitious specimens wrapped in composite materials of paste- and liquid-form epoxy resin as matrices, and unidimensional and two-dimensional carbon fiber fabrics for reinforcement. Variation of values, ±5%.

**Figure 14 polymers-12-02058-f014:**
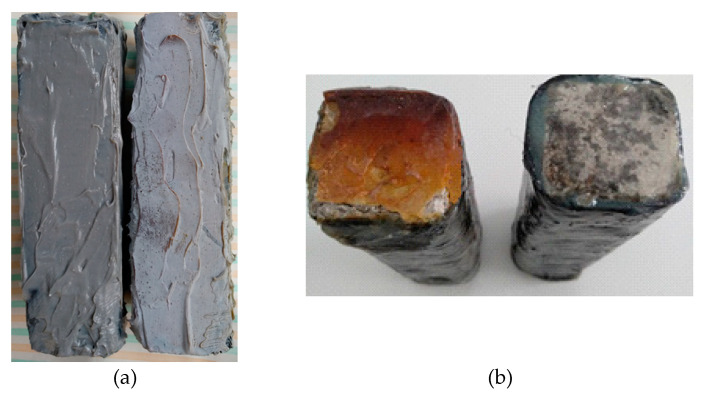
(**a**) Comparison between cement specimen with paste epoxy and carbon fiber sheet; (**b**) Comparison between cement specimen with liquid epoxy and carbon fiber sheet.

**Figure 15 polymers-12-02058-f015:**
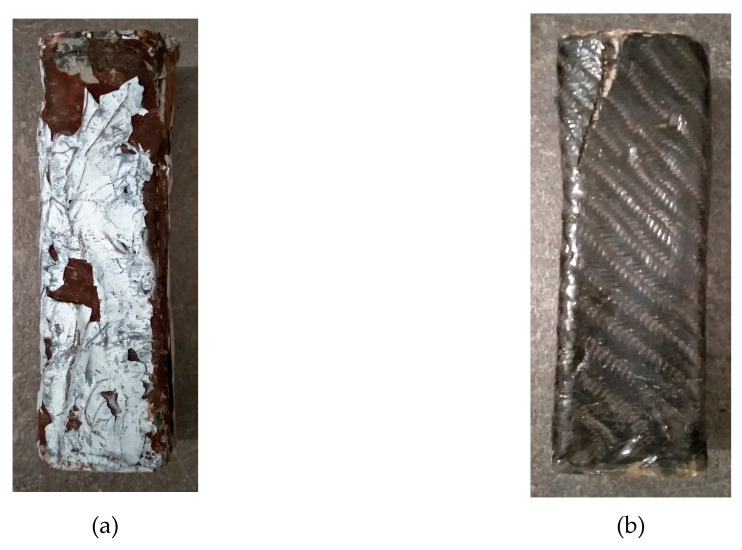
(**a**) Compressed cement specimen with paste epoxy and carbon fiber sheet; (**b**) Compressed cement specimen with liquid epoxy and carbon fiber sheet.

**Figure 16 polymers-12-02058-f016:**
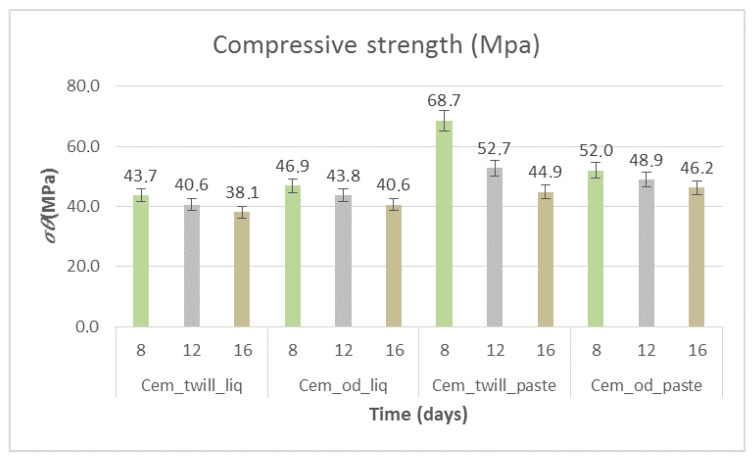
Compressive strength of cement specimens wrapped in composite materials of paste- and liquid-form epoxy resin as matrices and unidimensional and two-dimensional carbon fiber fabrics for reinforcement and subjected to heat treatment for artificial aging. Variation of values, ±5%.

**Figure 17 polymers-12-02058-f017:**
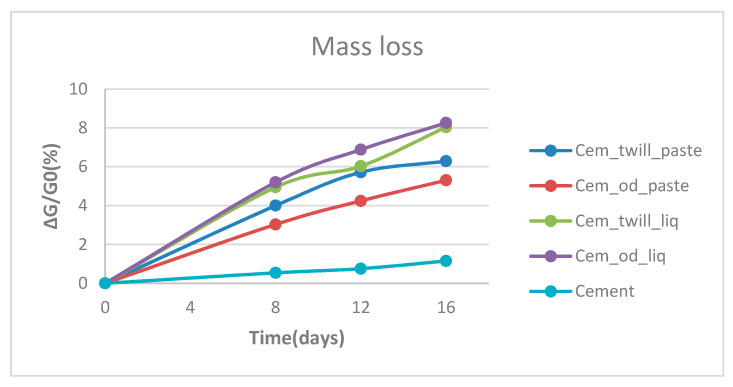
Thermal behavior of specimens subjected to artificial aging heat treatment for 16 days.

**Table 1 polymers-12-02058-t001:** Composite materials of epoxy resin–carbon fiber and cement specimens.

Composite Material Code	Cement (% *w/w*) (Cem)	Carbon Fibers (% *v/v*)		Epoxy Resin (% *w/w*)	
		Od	Twill	Liq	Paste
**Cem**	100	0	0	0	0
**Cem_od_paste**	80	10	0	0	10
**Cem_od_liq**	80	10	0	10	0
**Cem_twill_paste**	80	0	10	0	10
**Cem_twill_liq**	80	0	10	10	0
**Od_paste**	0	50	0	0	50
**Od_liq**	0	50	0	50	0
**Twill_paste**	0	0	50	0	50
**Twill_liq**	0	0	50	50	0

Notes: Od: one-dimension; Liq: liquid; Cem: cement.

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
