# Peer review of "Polymer Composite Materials Fiber-Reinforced for the Reinforcement/Repair of Concrete Structures"

_polymers, 2020, doi:10.3390/polym12092058_

Round 1

Reviewer 1 Report

This paper deals with the use of carbon fiber reinforced polymer matrix composites as a shear reinforcement of Portland cement mortars loaded in a three-point loading configuration to determine shear and flexural loads. The subject is important and it should be interesting to the composites scientific community.

There are several points that should be addressed before full acceptance of this paper.

  1. Was the heat treatment (aging) at 65-75 °C for 8, 12 and 16 days applied simultaneously to the UV exposure? Please explain.
  2. What faces of the cement flexural test specimens were wrapped with the composite materials? Which was the fiber direction of the composite on the cement specimens? Photographs of the specimens would highly enhance this manuscript.
  3. Specify the fiber direction of the compression test specimens and the number of faces coated with the composite material.
  4. It is assumed that only one layer of composite material was applied to all the specimens. Confirm.
  5. What was the failure mode of the composite and/or cement specimens? Were these failures as expected?
  6. Correct figure caption of figure 2. (number is not correct)
  7. Correct figure caption of figure 4. Please translate to English language.
  8. The authors state in page 5 of 7, line 140: “Based on Figure 5, loss of mass occurred during the artificial aging heat treatment in addition to the decrease in compressive strength. In the first days of exposure, the specimens emitted a strong smell of resin – clearly due to the degradation of the resin – which is the reason for the loss of mass….” Epoxy resins are usually very stable at the temperature interval they were subjected, but depending on the curing agent used. How can it be assured that the weight loss shown in figure 5 is not loss of water trapped in the cement mortar?

Author Response

Photos and diagrams were added to the article as suggested

  1. Was the heat treatment (aging) at 65-75 °C for 8, 12 and 16 days applied simultaneously to the UV exposure? Please explain.

We modified the corresponding text as “During the process of artificial aging the temperature in the chamber reaches the range 65-75οC.”

  1. What faces of the cement flexural test specimens were wrapped with the composite materials? Which was the fiber direction of the composite on the cement specimens? Photographs of the specimens would highly enhance this manuscript.

We modified the corresponding text as “The cement specimens were reinforced peripherally to their four sides (Figure 4a), with two layers of the composite material. The unidimensional carbon fiber fabric was used to wrap the cement specimen with the direction of the fibers placed vertically to the compression force. As for the two-dimension fabric, the one direction of its fibers was parallel and the other one vertical to the compression force, as shown in Figure 1.”

  1. Specify the fiber direction of the compression test specimens and the number of faces coated with the composite material.

We modified the corresponding text as “The cement specimens were reinforced peripherally to their four sides (Figure 4a), with two layers of the composite material. The unidimensional carbon fiber fabric was used to wrap the cement specimen with the direction of the fibers placed vertically to the compression force. As for the two-dimension fabric, the one direction of its fibers was parallel and the other one vertical to the compression force, as shown in Figure 1.”

  1. It is assumed that only one layer of composite material was applied to all the specimens. Confirm.

We modified the corresponding text as “The cement specimens were reinforced peripherally to their four sides (Figure 4a), with two layers of the composite material.”

  1. What was the failure mode of the composite and/or cement specimens? Were these failures as expected?

We modified the corresponding text as “As shown in Figure 12, the breaking of the specimens in all cases is as expected [15, 16], perpendicular to the dimension of the force exerted on them.”

  1. Correct figure caption of figure 2. (number is not correct)

  1. Correct figure caption of figure 4. Please translate to English language.

  1. The authors state in page 5 of 7, line 140: “Based on Figure 5, loss of mass occurred during the artificial aging heat treatment in addition to the decrease in compressive strength. In the first days of exposure, the specimens emitted a strong smell of resin – clearly due to the degradation of the resin – which is the reason for the loss of mass….” Epoxy resins are usually very stable at the temperature interval they were subjected, but depending on the curing agent used. How can it be assured that the weight loss shown in figure 5 is not loss of water trapped in the cement mortar?

We modified the corresponding text as “Based on Figure 17, loss of mass occurred during the artificial aging heat treatment in addition to the decrease in compressive strength. In the first days of exposure, the specimens emitted a strong smell of resin – clearly due to oxidation [17] and the evaporation of superfluity solvents and hardener. And that was the reason for the loss of mass. The cement specimen has the smallest loss of mass as expected [18].”

Reviewer 2 Report

See the attached document, which is the review for the Authors.

Author Response

Photos and diagrams were added to the article as suggested

  1. The tests must be sufficiently well explained that someone else knowledgeable about the field could repeat the experiments and the study. Therefore, sufficient information must be provided for a capable researcher to reproduce the experiments described. This criticism includes providing photos, diagrams of the tested specimens and set-up, together with dimensions and characteristics of the equipment. Ultimately, the Authors have to add additional figures that would add clarity to the text. The revised version resubmitted must provide the study’s statement of purpose. Moreover, all elements of the revised version resubmitted must relate logically to that study’s statement of purpose.

Accordingly, the revised version resubmitted must provide the figures and tables necessary to the understanding of the article and the conclusions. The lack of figures of the submitted manuscript compromises the article’s message. The results must be soundly interpreted and must related to existing knowledge on the topic. In the revised version resubmitted, all possible interpretations of the test results must be considered. As shown in the papers that are suggested below, there are alternative hypotheses that are consistent with the available data. Moreover the findings must be properly described in the context of the published literature. Another point is that the limitations of this study must be discussed, so that the major limitations are analyzed. Actually, the submitted material ignores the great effort that was spent on this subject in the last decades, and only a few important papers are cited. In the revised version resubmitted, appropriate references must be cited, where previously established methods are used. To that end, I suggest citing the following papers. P. Foraboschi. Effectiveness of novel methods to increase the FRP-masonry bond capacity. Composites Part B: Engineering, 2016; 107(December): 214- 232.

  1. Foraboschi. Analytical model to predict the lifetime of concrete members externally reinforced with FRP. Theoretical and Applied Fracture Mechanics, 2015; 75(1): 137-145. Although all research builds on prior results, noticeable similarity with previously published findings may reduce that originality which forms a crucial consideration in the prioritization of submissions for further review. Therefore, the revised version resubmitted must represent a conceptual advance over previously published work. In the revised version resubmitted, the conclusions must be sound and justified, and they must follow logically from data presented. SO, in the revised version resubmitted, write a set of conclusions, or a summary and conclusion, in which the significant implications of the information presented in the body of the manuscript are reviewed. The Authors have also to improve some major issues of the Section 3, to save the potential readers’ effort that I needed to invest in order to understand that part. Ultimately, the revised version resubmitted has to present a specific, easily identifiable advance in knowledge. Accordingly, the content and the results must be applicable and useful to the research and/or the profession. Which means that the claims in the revised version resubmitted must be sufficiently novel to warrant publication.

Photos and diagrams were added to the article as suggested

  • We modified the corresponding text as “This is especially important for the non-reinforced structures (concrete and brick masonry), which are the areas that undergo more stress. Since they constitute a very promising field, the cement matrix materials reinforced with composite materials have been widely studied over the last two decades [6, 7, 8, 9]. This is important since the use of brick masonry which carries force loads is not the standard in a number of countries. For example, in some countries, brick masonry is used only as a covering surface filling the space between concrete columns (hollow bricks). Otherwise, the reinforcement of brick masonry constitutes a specific sub-field that has been studied thoroughly yielding in that respect effective solutions [8] but none has been standardized to this day.”

  • We modified the corresponding text as “The fact that the concrete structures accept the loads vertically to the small surface of the structure makes necessary the reinforcement of the rest of the surfaces.. In particular, it is necessary to extend the life of concrete structures with the reinforcement of the main structure, not only of the columns but also of the concrete masonry (wall). The outer reinforcements of concrete structures are important in two ways:

It is without question that this type of reinforcements cannot remove the inherent inner imperfections as such which might lie hidden within these concrete structures – and at the same time, they are difficult to be identified [9]. Yet it is exactly because of these difficulties in total repairing that the outer reinforcement needs detailed study with the view of their role in extending the body lifespan and also delaying further inner debonding

Outer reinforcements protect the whole structure from the effects of external factors that contribute to their deterioration

The aim of this study is to contribute to the expansion of the experimental database on the subject of compression strength of cementitious structures using composite materials as external reinforcement. Additionally, it is essential to examine how the climate (artificial aging) affects the mechanical properties of the concrete which has been reinforced with composite material. To this purpose, cement specimens were manufactured according to EN 196-1: 1995 and were coated with carbon fiber fabrics (unidimensional and two-dimensional) and epoxy resin. Although it is used in complex shapes, the two-dimensional fabric is not common to utilize for the reinforcement of this type of structures. Additionally, composite materials were manufactured with epoxy resin as a matrix and carbon fiber fabrics as a reinforcement, to estimate their mechanical properties and, in particular, their flexural and shear strength. For comparison reasons, the compressive strength was estimated in cement samples, both reinforced and non-reinforced ones. Furthermore, cement samples reinforced with carbon fiber and epoxy resin were subjected to artificial aging to investigate to what extent the mechanical properties are affected over time in healthy but also vulnerable surfaces.”

  • We modified the corresponding text as “The cement specimens were reinforced peripherally to their four sides (Figure 4a), with two layers of the composite material. The unidimensional carbon fiber fabric was used to wrap the cement specimen with the direction of the fibers placed vertically to the compression force. As for the two-dimension fabric, the one direction of its fibers was parallel and the other one vertical to the compression force, as shown in Figure 1. Moreover, paste epoxy resin was easier to apply during the manufacture. After the cement specimens were coated with the composite material, specimens from all cases were selected and subjected to heat treatment (aging) at 65-75 °C for 8, 12 and 16 days, for comparison purposes. The compressive strength was measured in all cement specimens.”

  • We modified the corresponding text as “To measure their bending and shear strength, the composites were made with both liquid and paste resin with unidimensional and two-dimension (twill) fabric. All composites were manufactured using the hand-layup method [8]. Figure 4b shows the dimensions of the specimens for the bending and shear strength measurement. The calculation of carbon fiber fabric layers for the manufacture of the composite materials was in accordance to the thickness of the fabrics. For the unidimensional fabric, the thickness is 0,630 mm, and for the two-dimension (twill) fabric, the thickness is 0,285 mm. So to manufacture composite materials with 3 mm thickness: for the unidimensional fabric are required five (5) layers which weigh 358 g, and for the two-dimension (twill) fabric, eleven (11) layers which weight 405 g. Furthermore, to manufacture the composite materials in case of unidimensional fabric, the epoxy resin that was used weights 358 g and for the two-dimension (twill) fabric 405 g. The specimens that were manufactured in the present work are depicted in Figure 5a. Those specimens suitable for the mechanical tests are shown in Figure 5b.”

  • We modified the corresponding text as “Figure 9 shows specimens that were tested in bending and shear strength. All specimens (of all cases) that were tested in shear strength they broke completely. In contrast, those that were tested in bending strength did bend, but they did not break. Figure 10 shows the bending strength and Figure 11 shows the shear strength of the composite materials with paste and liquid epoxy resin as matrix and reinforced with unidimensional and two-dimension carbon fiber fabrics. It is apparent from Figure 10 that the composite materials with unidimensional (Od) carbon fabrics outperform in bending strength the two-dimension (Twill) carbon fabrics by ~30%, As for the matrix, the liquid form epoxy resin exhibits greater bending strength which is 405.13 Mpa. As shown in Figure 11 the shear strength is approximately the same in all cases at around 30 MPa. Specifically, the specimens that were reinforced with unidirectional fabrics exhibit slight higher shear strength (33 MPa).”

  • We modified the corresponding text as “This paper presents an experimental and theoretical investigation on the compression behavior of the reinforced small-scale cement beams with CFRP reinforcement after they have been heat-treated (artificial aging). The main findings are drawn as follows:
  • Unidimensional carbon fiber fabrics as reinforcement provide better bending strength than those of the two-dimension with twill woven ones, in all cases, independent of the form of the epoxy matrix.
  • The maximum flexural strength is found in the composite material of liquid epoxy resin reinforced with unidimensional carbon fiber fabric, due to the better penetration of the resin into the fibers.
  • It is found that the shear strength of composite materials does not change significantly depending on the case, with the cases of composite materials reinforced with unidimensional fabrics being slightly higher in strength.
  • An increase in compressive strength of 41.93% was observed in the case of a cement specimen reinforced with composite material of paste epoxy resin with a two-dimension carbon fiber fabric (Cem_twill_paste). While the choice of the twill pattern showed indeed better results, the role of the matrix of the composite material cannot be overstated.
  • Also, it was observed that paste epoxy resin was easier to apply to the specimen with carbon fabrics. However, it is worth noting that the two-dimension carbon fiber fabric adhered better to the specimen.
  • Artificial aging, the procedure of heat treatment, shows a significant reduction in the mechanical properties of composite materials of up to 40%. This is also evident in paste resin specimens where, by the end of artificial aging, the resin had created a layer of dust on the surface of the specimens. This indicates the oxidation and the evaporation of superfluity solvents and the hardener and, therefore, the loss of mass.
  • The composite materials with life-extending anti-UV epoxy resins are expected to maintain their high mechanical properties.

This study clearly reveals that the composite material of paste epoxy resin as a matrix and twill fabric as reinforcement has better results in reinforcing cementitious specimens. Of all the specimens that were subjected to artificial aging the aforementioned ones proved to exhibit the best mechanical properties.”

Round 2

Reviewer 1 Report

  1. The purpose of using a composite material for retrofitting of a cement based structure is provide a reinforcement in directions of stress components that the cement structure cannot support. What is the benefit of wrapping all the surfaces of the specimens? Which stress component will be taken by the coposite material?
  2. Please specify the span lengths between the supports for both the bending and shear tests. What was the speed of loading used?
  3. The authors state that "the evaporation of superfluity solvents and the hardener and, therefore, the loss of mass." What solvents are present in the epoxy? Is there a possibility that the lost mass is simply water evaporation? Please explain.
  4. Do you have any idea of the final resin/fiber volume ratio of the resulting composite materials?
  5. The authors state: "Figure 9 shows specimens that were tested in bending and shear strength. All specimens (of all cases) that were tested in shear strength they broke completely. In contrast, those that were tested in bending strength did bend, but they did not break".  One of the difficulties encountered in the 3-point loading mode is the separation of the bending and shears stress components,especially if the span length between supports is not selected correctly. Question: Is the failure mode observed the one expected for bending and shear ?

Author Response

  1. The purpose of using a composite material for retrofitting of a cement based structure is provide a reinforcement in directions of stress components that the cement structure cannot support. What is the benefit of wrapping all the surfaces of the specimens? Which stress component will be taken by the composite material?

This reinforcement method was chosen in order to increase the shear capacity of the concrete columns, to enhance their ductility and to improve the confinement level of concrete during mechanical stress [13, 14].

  1. Please specify the span lengths between the supports for both the bending and shear tests. What was the speed of loading used?

Based on the standards for overall bending tests, support span should be 16 (tolerance ±) times the thickness of the specimen (in this work 5 cm) and the speed no higher than 2 mm/min. For the shear tests the support span should be 8 (tolerance ±) times the thickness (in this work 2 cm) and the speed no higher than 2 mm/min.

applying pressure (1 mm/min).

  1. The authors state that "the evaporation of superfluity solvents and the hardener and, therefore, the loss of mass." What solvents are present in the epoxy? Is there a possibility that the lost mass is simply water evaporation? Please explain.

In the first days of exposure, the resin on the specimens emitted a strong smell of volatiles due to the evaporation of superfluity solvents i.e. MEK (Methyl Ethyl Ketone) that is present in the hardener of the resin. Those volatiles constitute the 3% up to 6% of mass loss and the remaining (~2%) is water from the cement as is it can be observed in Figure 17. The cement specimen has the smallest loss of mass, due to the water evaporation, as expected [20].

  1. Do you have any idea of the final resin/fiber volume ratio of the resulting composite materials?

The volume ratio fiber/resin for the composite material is 53% v/v and for the reinforced cement specimens the fiber volume ratio is 3% v/v (for two layers carbon fabric).

  1. The authors state: "Figure 9 shows specimens that were tested in bending and shear strength. All specimens (of all cases) that were tested in shear strength they broke completely. In contrast, those that were tested in bending strength did bend, but they did not break". One of the difficulties encountered in the 3-point loading mode is the separation of the bending and shears stress components, especially if the span length between supports is not selected correctly. Question: Is the failure mode observed the one expected for bending and shear?

This is because in bending strength the fabric receives the load preventing the specimen from breaking.

Reviewer 2 Report

I recommend that the revised version of the article that has been resubmitted is accepted and published.

The Authors have considered how I had commented their article and have suitably and carefully addressed all my comments.

Now, the article adds to the subject and the presentation saves the readers’ effort to understand the article.

Moreover, in the revised version resubmitted, the appropriate structure and language have been used and the presentation is good and consistent, now. In particular, the description of the new methodology is accurate and clear.

Author Response

Dear Sir or Madam,

We would like to thank you for your time and effort in order to bring the paper in a most fidelitous form for publication.

We are at your disposal for providing you with more information in case it is needed.

Yours truly,

Round 3

Reviewer 1 Report

I consider that this paper can be recommended for publication in its present form.